# Pharmacokinetics, Prostate Distribution and Metabolic Characteristics of Four Representative Flavones after Oral Administration of the Aerial Part of *Glycyrrhiza uralensis* in Rats

**DOI:** 10.3390/molecules27103245

**Published:** 2022-05-19

**Authors:** Haifan Liu, Guanhua Chang, Wenquan Wang, Zuen Ji, Jie Cui, Yifeng Peng

**Affiliations:** 1Institute of Medicinal Plant Development, Chinese Academy of Medical Sciences & Peking Union Medical College, Beijing 100193, China; 15530156755@163.com (H.L.); jcui@implad.ac.cn (J.C.); 2Beijing Wehand-Bio Pharmaceutical Co., Ltd., Beijing 102629, China; y13194305682@163.com; 3Engineering Research Center of Good Agricultural Practice for Chinese Crude Drugs, Ministry of Education, Beijing 100102, China; 4Xinjiang Key Laboratory for Reserch of Licorice and Products, Korla 841011, China; chinn203@126.com; 5Xinjiang Quanan Pharmaceutical Co., Ltd., Korla 841011, China

**Keywords:** aerial parts of *Glycyrrhiza uralensis*, flavonoids, pharmacokinetics, prostate distribution, metabolic characteristics

## Abstract

(1) Background: The aerial part of *G. uralensis* had pharmacological effects against chronic non-bacterial prostatitis (CNP), and flavonoids are the main efficacy components. The purpose of this study was to obtain the pharmacokinetics, prostate distribution and metabolic characteristics of some flavonoids in rats. (2) Methods: The prototype flavones and the metabolites of four representative flavonoids, namely puerarin, luteolin, kaempferol and pinocembrin in plasma, prostate, urine and feces of rats were analyzed by UPLC-Q-Exactive Orbitrap-MS. In addition, the pharmacokinetic parameters in plasma and distribution of prostate of four components were analyzed by HPLC-MS/MS. (3) Results: In total, 22, 17, 22 and 11 prototype flavones were detected in the prostate, plasma, urine and feces, respectively. The metabolites of puerarin in the prostate are hydrolysis and glucose-conjugated products, the metabolites of kaempferol and luteolin in the prostate are methylation and glucuronidation, and the metabolites of pinocembrin in the prostate are naringenin, oxidation, sulfation, methylation and glucuronidation products. The t_1/2_ of puerarin, luteolin, kaempferol and pinocembrin was 6.43 ± 0.20, 31.08 ± 1.17, 18.98 ± 1.46 and 13.18 ± 0.72 h, respectively. The concentrations of the four flavonoids in prostate were ranked as kaempferol > pinocembrin > luteolin > puerarin. (4) Conclusions: Methylation and glucuronidation metabolites were the main metabolites detected in the prostate. A sensitive and validated HPLC–MS/MS method for simultaneous determination of puerarin, luteolin, kaempferol and pinocembrin in rat plasma and prostate was described, and it was successfully applied to the pharmacokinetic and prostate distribution studies.

## 1. Introduction

The proportion of total flavonoids in the dry stem and leaves of *Glycyrrhiza uralensis* Fisch. (licorice, *G. uralensis*) is up to 5.64% [1]. We have reported previously that the aerial parts of *G. uralensis* exhibit antioxidant [2], antibacterial [3], anti-inflammatory [2], antitumor [4] and antidiabetic [5] properties. A research group showed that licorice can partly confer an antichronic nonbacterial prostatitis (CNP) effect [1]. To our knowledge, despite the widespread use of the aerial parts of *G. uralensis* and extensive studies on the traditional drug sites of *G. uralensis*, the metabolites, pharmacokinetics, prostate distribution and metabolic pathways of flavonoids of the aerial parts of *G. uralensis* in vivo are unknown. In addition, most studies have identified the metabolic components in plasma and urine, whereas the flavonoid components in prostate have been understudied. Therefore, determining the metabolites, pharmacokinetics, prostate distribution and metabolic pathways of the flavonoids in aerial parts of *G. uralensis* is of immense significance for better understanding the mechanism and clinical application of the aerial parts of *G. uralensis* against CNP.

Over the past few decades, liquid chromatography-mass spectrometry has become a powerful tool for the in vitro identification of chemical components [5,6]. Ultra-performance liquid chromatography plus Q-Exactive Orbitrap tandem mass spectroscopy (UPLC-Q-Exactive Orbitrap-MS) can help in determining high-resolution and accurate mass numbers, as well as multilevels of fragmentation, and it comprises various data processing software to perform rapid analysis of complex material systems [7]. In this study, we confirmed the molecular composition of the relevant matter and its characteristic fragment ions based on the molecular mass data obtained through high-resolution MS. Based on the relevant literature and reference standards, the main protoflavones and the metabolites of four representative flavones in biological samples were analyzed and identified. A selective and sensitive HPLC–MS/MS method was first developed and validated for the determination of puerarin, luteolin, kaempferol and pinocembrin in rat plasma and prostate. The method was successfully applied to evaluate the pharmacokinetics of four flavones after oral administration of aerial parts of *G. uralensis* to rats. 

In this study, 22, 17, 22 and 11 prototype components were detected in the prostate, plasma, urine and feces samples, respectively. The pharmacokinetic parameters, prostate distribution and metabolic characteristics of four representative flavones, namely puerarin, luteolin, kaempferol and pinocembrin, were expounded, which provides a scientific basis for clarifying the pharmacodynamic characteristics and further development and utilization.

## 2. Results

### 2.1. Characterization of Prototype Flavonoids in Biological Sample

To characterize the chemical constituents of the biological sample, all known compounds were identified by comparing them with reference standards. For unknown compounds, the accurate mass values of selected compounds were observed and compared with their exact mass (theoretical mass) values manually; the structures were tentatively characterized based on their chromatographic data, referring to previous literature. By comparing ion chromatograms, peaks that appeared both in the dosed rat biological sample but not in the blank rat sample were considered to be the components absorbed into the biological sample. By comparing these components with medicinal materials [8], common components were considered to be the components absorbed into the biological sample. Finally, thirty-four protoflavones were identified in the biological sample. The distribution of protoflavones in biological samples is shown in Table 1. Spectra of 16 components in the reference solution and sample are shown in Figure 1.

Five prototype components (retrochalcone, kaempferol, luteolin, diosmetin and pinocembrin) in plasma, prostate, urine and feces can be found by analyzing the Wayne diagram of the prototype components in plasma, prostate, urine and feces (Figure 2); plasma, urine and feces had no other common components except the above five components; plasma and urine share only one prototype component (isorhamnetin); there are seven prototype components (astragalin, schaftoside, liquiritin, genistein, ononin, isoliquiritin and wogonin) in plasma and feces. The common prototype that is distributed in plasma and urine or plasma and feces is the component that is both excreted in urine and feces after plasma absorption and reabsorption in the kidney and gut. The prototypes detected in urine and feces are mainly flavonoids, flavonols and dihydroflavonoids. The dihydroflavanoid glycosides were excreted by feces.

From the perspective of flavonoid glycosides and flavonoid aglycones distribution (Figure 3), there are a large number of flavonoid glycosides in prostate and feces, a large number of flavonoid aglycones in urine, and the same number of flavonoid glycosides and aglycones in plasma.

The results of this paper have shown that the amount of flavonoid glycosides and aglycones was different in biological samples. Studies have shown that the absorption of flavonoid glycosides and flavonoid aglycones is quite different. Generally, the absorption of flavonoid aglycones is better than that of flavonoid glycosides. The liposolubility and molecular spatial structure of flavonoid aglycones are small, so that flavonoid aglycones can be directly absorbed by the passive diffusion of villi epithelial cells on the small intestinal wall. It is generally believed that aglycones are mainly absorbed through a simple diffusion [20,21]. Flavonoid glycosides have high hydrophilicity and relatively large molecular weight, which are generally considered to be difficult to be absorbed in the small intestine. In addition, flavonoids are weakly acidic and weakly alkaline, so they have poor solubility and poor absorption in the gastrointestinal tract, belonging to a class of compounds with poor absorption [22].

According to the method described above, 17 and 22 identified compounds were absorbed into rat plasma and prostate in prototype. They can be speculated to perhaps be the primary active components of partial anti-CNP. 

### 2.2. Metabolism of Four Representative Flavonoids

Four representative flavonoids (puerarin, luteolin, kaempferol and pinocembrin) of the aerial part of *G. uralensis* were selected to identify their metabolites in rats (Figure 4). These four components are considered to be pharmacodynamic ingredients of anti-CNP. The plasma, urine, feces and prostate samples were analyzed by UPLC-Q-Exactive Orbitrap-MS to characterize the metabolites, and high-resolution mass spectra of the metabolites were obtained to establish their molecular formulas (Table 2). The structures were identified by comparing them with reference standards and previous literature. 

#### 2.2.1. Study on the Metabolism of Puerarin

Identification of puerarin

The molecular ion peak of puerarin was *m*/*z* 415.1028 [M − H]^−^ and its molecular formula is C_21_H_20_O_9_. The secondary mass spectrum of puerarin contained *m*/*z* 295, 277, 267 and 253 fragment ion information. The loss of the C_4_H_8_O_4_ group *(m*/*z* 120) resulted in the formation of a fragment ion of *m*/*z* 295, representing the fragmentation of sugar molecules in glycosides, which further lost CO and H_2_O to produce the fragment ions of *m*/*z* 267 and *m*/*z* 277, respectively. The fragment at *m*/*z* 253 indicated that one molecule of glucose was removed, and glycosidic bonds in isoflavones were easily disrupted, resulting in the C ring cleavage of carbonyl, loss of CO at the C4 position and generation of the [^1,3^A]^−^ fragment ions through Retro Diels–Alder (RDA) cleavage. The cleavage pathway is shown in Figure 5.

2.Metabolism of puerarin

After oral administration, only puerarin and methyl-daidzein-glucoside were detected in plasma. Our study found that puerarin was mainly excreted through the urine, and in addition to the prototype component, methylated products, glucose conjugates and glucuronide conjugates were also detected in the excreta, and this study is the first finding that 1, 1-M2, 1-M4 and 1-M6 can enter prostate tissue and are presumed to be the active ingredients of anti-CNP. Its metabolic pathway is shown in Figure 6.

#### 2.2.2. Study on the Metabolism of Kaempferol

Identification of kaempferol

The molecular ion peak of kaempferol was *m*/*z* 285.04 [M − H]^−^ and the molecular formula is C_15_H_10_O_6_. The obtained secondary mass spectrum contained the fragment ion information of *m*/*z* 135, 137, 151, 241 and 257; the cleavage pathway is shown in Figure 7.

2.Metabolism of kaempferol

It has been shown that glucuronide conjugates, methylated products and sulfate-bound metabolites of kaempferol can be detected in plasma and urine.

In addition to the conjugated forms, kaempferol also underwent oxidative conversion to quercetin, which was further subject to methylation, yielding isorhamnetin. Both quercetin and isorhamnetin could also be glucuronidated. 

This study is the first to explore the metabolism of kaempferol in feces and prostate. The results show that it is mainly excreted in the form of prototype, quercetin, sulfur, methylation and glucuronic acid conjugates in feces. The metabolites of kaempferol in the prostate are mainly prototype, methylation and glucuronic acid conjugates. These metabolites may play a role in prostate disease. The metabolic pathway of kaempferol is shown in Figure 8.

#### 2.2.3. Study on the Metabolism of Luteolin

Identification of luteolin

The molecular ion peak of luteolin was *m*/*z* 285 [M − H]^−^ and its molecular formula is C_15_H_10_O_6_. The obtained secondary mass spectrum contained fragment ions of *m*/*z* 151, 175, 257 and 267. The cleavage pathway is shown in Figure 9.

2.Metabolism of luteolin

After oral administration, luteolin-3′-glucuronide (Lut-3′-G), luteolin-4′-glucuronide (Lut-4′-G) and Lut-7-G have been identified in the biological specimen and Lut-4′-G and Lut-7-G were first detected in the prostate. 

In addition to detecting phase II metabolites, oxidation products (M6) were also detected. This study found that all the metabolites of luteolin can be excreted by urine. The results show that the methylation of luteolin, luteolin and glucuronidation products detected in the prostate may have some role in prostate disease. The metabolic pathway of luteolin is shown in Figure 10.

#### 2.2.4. Study on the Metabolism of Pinocembrin

Identification of pinocembrin

The molecular ion peak of pinocembrin was *m*/*z* 257 [M+H]^+^ and the molecular formula is C_15_H_12_O_4_. The obtained secondary mass spectrum contained fragment ions of *m*/*z* 97, 103, 131 and 153. The cleavage pathway is shown in Figure 11.

2.Metabolism of pinocembrin

After oral administration, hydroxylation and glucuronidation products (4-M1 and 4-M6) were detected in plasma, prostate, urine and feces. The M4 sulfate conjugates and the M5 methyl conjugates were also preliminarily speculated upon. Compounds M4 produced fragment ions with *m*/*z* of [M − H-SO_3_]^−^ (SO_3_-sulfate group). According to its molecular formulas, it was identified as the sulfated product of pinocembrin. For compound M5, the molecular ion peak *m*/*z* was 269.08 [M − H]^−^ and the chemical formula was C_16_H_14_O_4_; its molecular weight increased by 14 Da compared with that of pinocembrin. The fragment ion peak was *m*/*z* 254 [M-H-CH_3_]^−^ (CH_3_-methyl group), indicating that the compound contains a CH_3_ molecule, and hence, we speculated that it is the methylation product of pinocembrin. 

Metabolites 4, 4-M1, 4-M2, 4-M4, 4-M5, 4-M6 and 4-M7, respectively, were detected in the prostate; they are pinocembrin, naringenin, sulfation, methylation, glucuronidation and oxidation products, and it was the first time that a combination of sulfate and naringenin was found in the prostate. These metabolites all contain OH-5 groups. The metabolic pathway of pinocembrin is shown in Figure 12.

### 2.3. Method Validation on Pharmacokinetics and Prostate Distribution

#### 2.3.1. Selectivity

The typical chromatograms obtained from the control plasma/prostate homogenates, the control samples spiked with analytes and IS, the samples of rat plasma/prostate homogenates obtained 4 h after oral administration of aqueous extract of the aerial part of *G. uralensis*, as well as the mixture of analytes and IS solution, respectively, are illustrated in Figure 13 and Figure 14. No endogenous interference was observed.

#### 2.3.2. Calibration Curves

The equation and correlation coefficient (*r^2^*) of the samples of plasma/prostate homogenates are shown in Table 3.

#### 2.3.3. Matrix Effect and Extraction Recovery

Extraction recovery ranged from 80.488% to 104.914% for compounds in rat plasma and prostate (Table 4). Matrix effects ranged from 81.795% to 109.639% for compounds. The results showed no significant ion inhibition or enhancement that occurs in this method, thus indicating a negligible matrix effect on the ionization of the analytes.

#### 2.3.4. Accuracy and Precision

The intra-day and inter-day precision and accuracy were <20% for both plasma and prostate (Table 5). 

#### 2.3.5. Stability

Stability data are summarized in Table 6 and they indicated that the four analytes in plasma and prostate were stable for 24 h in autosampler condition after preparation, for at least three freeze-thaw cycles. 

### 2.4. Pharmacokinetics Analysis of Four Representative Flavonoids 

The validated method was successfully applied to monitoring the concentrations and pharmacokinetic studies of four flavonoids in rat plasma and prostate after oral administration of the aerial part of *G. uralensis* extract (1 g/kg) every day for 31 days. The mean plasma concentration–time curves (*n* = 6) of puerarin, luteolin, kaempferol and pinocembrin are shown in Figure 15. The pharmacokinetic parameters are listed in Table 7.

The time to reach the maximum plasma concentration (T_max_) was 0.50 ± 0.04 h for puerarin, 0.87 ± 0.05 h for luteolin, 4.00 ± 0.17 h for kaempferol and 1.50 ± 0.05 h for pinocembrin. The elimination half-time (t_1/2_) of puerarin, luteolin, kaempferol and pinocembrin was 6.43 ± 0.20, 31.08 ± 1.17, 18.98 ± 1.46 and 13.18 ± 0.72 h, respectively. It can be inferred that the extract could have an effect on the absorption of four flavonoids in rat plasma. 

The results of the concentration–time curve showed that the blood concentrations of four flavones in rats increased to varying degrees in 1, 3, 5, 9, 11, 13, 15, 17, 19, 21, 25, 27, 29 and 31 days, suggesting that the effective components had a certain accumulation in rats. Puerarin, luteolin, kaempferol and pinocembrin reached steady state at 19, 9, 11, 5 and 5 days, respectively, which laid the foundation for the determination of the days of administration. At the same time, two or even three peaks appeared in the concentration–time curves of the four components after administration, indicating that four flavonoids in vivo may exhibit hepatoenteric circulation and gastric emptying.

### 2.5. Prostate Distribution of Four Representative Flavonoids 

The concentration in prostate showed a changing trend at the time points of 0.5, 1.5, 2.5, 6 and 8 h after oral administration of the aerial part of *G. uralensis* (1 g/kg) at 31 days (as shown in Figure 16).

According to the concentration histograms of the four components at different time intervals in the prostate, the drug concentrations of the three flavonoid aglycones (luteolin, kaempferol and pinocembrin) in the prostate were significantly higher than that of one flavonoid aglycone (puerarin), and the concentration of kaempferol in the prostate was particularly high, while the concentrations of pinocembrin, luteolin and puerarin decreased gradually. The concentrations of the four flavonoids were ranked as kaempferol > pinocembrin > luteolin > puerarin. This may be due to the low molecular weight of flavonoid aglycones, which are more likely to enter the prostate through the prostate membrane.

## 3. Discussion

### 3.1. Characterization of Prototype Flavonoids in Biological Sample

In this experiment, 9 and 12 flavonoid glycosides were detected in plasma and prostate respectively, but it was reported that flavonoid glycosides were not easily absorbed into plasma in prototype form. However, nine flavonoid glycosides were detected in plasma in this experiment, namely astragalin, hyperoside, isoquercitrin, liquiritin, ononin, puerarin, rutin, schaftoside and vitexin. The reason for analysis is that there are many components in the aerial parts of *Glycyrrhiza uralensis Fisch* [33]. The method adopted in this paper is to detect the components in the mixed biological samples collected in different time periods, and these components are absorbed into the blood at different times. Because this experiment is qualitative analysis, the concentrations of all components in the biological samples are not detected, and it is speculated that the concentrations of these components in rats are low. Most of them are metabolized into other components and absorbed. For example, Dong Lei [34] et al. showed that after the administration of flavonoid glycoside (3′,4′- dimethoxyflavonol-D-glucoside) by gavage, the concentrations of prototype components and glucuronic acid conjugates in plasma were detected. The results showed that both prototype components and metabolites were detected in plasma, but the concentrations of metabolites were significantly higher than those of prototype components, and flavonol glycosides were rapidly and widely distributed in various tissues after gavage. In addition, there are also reports that flavonoid glycosides were detected in plasma after oral administration of traditional Chinese medicine extract. For example, studies have shown that isoquercitrin was detected in rabbit plasma after oral administration of Qikui sustained-release tablets [35], isoquercitrin, rutin and astragalin prototype components were detected in rat plasma after oral administration of mulberry leaf extract [36], and, isoquercitrin, hyperoside and quercitrin prototype components were detected in rat plasma after oral administration of *Apocynum venetum* leaf extract [37]. The prototype component of isoliquiritin was detected in the blood of rats after oral administration of Banxia Xiexin Decoction [38], and ononin was detected in the plasma of rats after oral administration of *Trifolium pratense* extract [39].

The study results showed that nine components of the prostate (sorhamnetin-3-O-rutinoside, baicalin, diosmin, isoschaftoside, isovitexin, puerarin, calycosin, isoliquiritigenin and quercetin) were not detected in the plasma and they are mainly the flavonoid glycosides. The speculated reason is that some components are not absorbed or some components that are hydrolyzed in the gastrointestinal tract are absorbed into the blood, and then the prototype components of the resynthesis are distributed in the prostate tissue to play a role. For example, baicalin is not easy to be absorbed into blood. It is first hydrolyzed into baicalein in vivo, and then baicalin is regenerated by glucuronidation to play a role [40,41,42,43].

The dihydroflavanoid glycosides were excreted by feces. Analyzing the reasons, after oral administration, most Chinese medicines were excreted in the form of hydrophilic prototypes, or metabolized into more polar molecules through the liver, and then excreted by urine. When passing through the digestive tract, due to the role of gastric acid, digestive enzymes and intestinal flora, a variety of metabolic reactions may occur, so that some drugs are inactivated in the intestines and the number of prototype drugs absorbed into the body is reduced. Most drugs are mainly absorbed in the small intestine. A large number of enzymes and intestinal flora in the small intestine metabolize glycoside components to a certain extent and convert glycoside components into aglycones. However, due to the low bioavailability of dihydroflavanoside, it is mainly excreted by feces [44,45,46].

There are more flavonoid glycosides in prostate and feces and they are mainly composed of flavonoid glycosides and flavonol glycosides. Studies have shown that some flavonoids are thought to have beneficial health effects, such as preventing atherosclerosis and certain types of cancer [47,48]. For example, baicalin is one of the main bioactive flavone glucuronides and current studies have shown various strong pharmacological properties of baicalin, such as antioxidative, antiviral, anti-inflammatory, antitumor, antiradical, antiproliferative, cardioprotective, and so on. There are some reports on the absorption of flavonoids in the human body; Izumi [49] concluded that soybean isoflavones of aglycone type absorbed faster and more than the glycosidic type in the human body. This conclusion is the same as the rule of rats in this experiment.

At the same time, in addition to the beneficial ingredients, some studies have shown that the aerial part of *Glycyrrhiza uralensis* also has some toxicity. Zhao’s [50] study showed that in the 30-day repeated oral safety test based on the inflammatory model, the water extract and alcohol extract of the aerial part of *Glycyrrhiza uralensis* had potential toxicity, and the rats had liver and kidney lesions. Serum globulin and serum creatinine were significantly increased, while liver index, kidney index and heart absolute weight were significantly increased. Studies have shown that some flavonoids also have certain toxicity; excessive quercetin and kaempferol can produce toxicity to mouse McCoy cell lines. As the B ring of kaempferol has one hydroxyl less than quercetin, it has higher lipophilicity and is more likely to destroy the cell membrane and produce cytotoxicity [51]. Studies have shown that excessive naringenin may distort or even kill *Bufo arenarum* embryos [52]. Mitsuyoshi [53] found that high concentrations of flavonoid (luteolin) and flavonols (quercetin, rutin) had toxic effects on normal cells—human embryonic lung fibroblasts (TIG-1) and human umbilical vascular endothelial cells (HUVE) in a dose-dependent manner. Walle et al. [54] confirmed that although quercetin has been widely regarded as an antioxidant that can scavenge reactive oxygen species in cells, studies have shown that reactive oxygen species can metabolize flavonoids and produce products that interfere with the physiological roles of key macromolecular compounds in the body [54]. The results of this study found that the prostate contains quercetin, kaempferol, naringenin, luteolin and rutin, so we should pay attention to this in later toxicological studies. Previous studies have shown that the toxic and side effects of flavonoids are created when the drug is overused. However, because there are many types of flavonoids in traditional Chinese medicine, and some components are unknown, its contents still need to be further studied.

### 3.2. Metabolism of Four Representative Flavonoids

#### 3.2.1. Metabolism of Puerarin

After oral administration, only puerarin and methyl-daidzein-glucoside were detected in plasma, consistent with the literature that reported that fewer metabolites of puerin were found in plasma [33]. 

Prasain et al. [55] suggested that puerarin is absorbed from the gastrointestinal tract without being hydrolyzed, and one of the routes of excretion of puerarin could be through the bile into the intestine and hence to the colon, where it is likely to be partly metabolized by colonic microflora and then reabsorbed. 

Puerarin belongs to isoflavones. Isoflavones have been reported to bind to the two isotypes of estrogen receptor (ER), alpha (ERα) and beta (ERβ), and to exhibit estrogenic or anti-estrogenic properties [56]. It can provide a reference for subsequent pharmacogenesis and efficacy research of anti-CNP.

In this study, 1-M3 (methylated metabolite) was not detected in urine, which was consistent with the results of fewer isoflavone methylated metabolites in urine [57]. In recent years, it has been found that isoflavones are mainly glucuronic-acid-binding metabolites in vivo, and there are also sulfated and both-binding metabolites, which is consistent with the results obtained in the previous experiment [58]. For example, Wang et al. [59] used UHPLC/Q-TOF MS to study the pharmacokinetic parameters of prototype and metabolites in rat plasma after oral administration of irisquinone. The results showed that the main metabolites of irisquinone glucuronidation and/or sulfation were found.

#### 3.2.2. Metabolism of Kaempferol

Hepatic and intestinal conjugation with subsequent excretion of phase II conjugates has been shown to be an important component of first-pass metabolism for many flavonoids [60,61]. Once absorbed, kaempferol is rapidly metabolized in the liver to form glucuronide, methyl and sulfate metabolites, which can be detected in the blood and urine. 

Quercetin and isorhamnetin could be glucuronidated. This metabolic route, however, appears to be uncommon in humans since no such metabolism is reported to occur in humans [62,63].

The results show that kaempferol was mainly excreted in the form of prototype, quercetin, sulfur, methylation and glucuronic acid conjugates in feces. The observation of Marsh et al. [64] shows that the intestinal contents show high β-glucuronidase activity. The kinetic constants obtained from the in vitro metabolism reveal that rates of glucuronidation are much higher than that for phase I oxidative metabolism [27].

#### 3.2.3. Metabolism of Luteolin

This study found that the metabolites of luteolin were mainly phase II metabolites. Lu et al. [65] selected the liver to study the metabolism of luteolin, an active component from chrysanthemum extract (CME). By using the primary liver cell model, it was concluded that the elimination rate of luteolin was 91.9%, and there was saturation in the elimination. Phase II metabolites accounted for 54% of the dosage. A small amount of luteolin could be converted into apigenin in vivo, and the two also had mutual inhibition. 

This study found that all the metabolites of luteolin can be excreted by urine. Aida Serra et al. [66] showed that the phenolic metabolites were uniformly distributed in all the tested tissues with a high concentration, showing preferential renal uptake of the metabolites, as observed by D’Angelo et al. [67] using intravenously injected [14C] hydroxytyrosol. 

The metabolism could probably be related to the ingested dose. High doses could saturate the conjugation metabolism of the phenolic compounds and this may allow the detection of free forms in the plasma in the present study, these probably being absorbed by passive diffusion.

#### 3.2.4. Metabolism of Pinocembrin

After oral administration, hydroxylation and glucuronidation products (4-M1 and 4-M6) were detected in plasma, prostate, urine and feces. LI Yuanyuan [32] found that the metabolic pathway of pinocembrin in bile, urine and feces is mainly hydroxylation and glucuronidation through the comparison of metabolites in vitro and in vivo in rats, and glucuronidation in the gastrointestinal tract and liver is the main method of its metabolic inactivation [32].

In this experiment, all the metabolites detected in the prostate contain an OH-5; the presence of an OH-5 reportedly provides for the much higher affinity and selectivity of genistein for ERβ compared to daidzein (which lacks the OH-5) [68,69].

#### 3.2.5. Metabolism of Flavonoids

The biological activity of flavonoid glycosides is often enhanced after metabolic transformation. Rutin is a common active flavonoid glycoside, but its poor absorption and low oral bioavailability limit its clinical application. It was found that rutin was metabolized to produce 3,4-dihydroxyphenylacetic acid which was easily absorbed into blood, and its activity was better than rutin. Some studies have also shown that liquiritin and isoliquiritin, the conversion products of intestinal flora of flavonoids in the Chinese herbal compound Huangqin Decoction, may be the pharmacodynamic substances that play a hepatoprotective role in vivo.

The main metabolic pathways of flavonoids with different structures are also different. For example, icariin is mainly hydrolyzed by glycosylation, while baicalin is hydrolyzed by glycosylation first and then generates glucuronide products, including baicalin prototype. Wen et al. [70] studied the differences in absorption characteristics of methylated and unmethylated flavonoids. The results showed that compared with unmethylated flavonoids, the apparent permeability coefficient of methylated flavonoids from the top side to the bottom side was higher, and it was 4–7 times that of unmethylated flavonoids. Although sulfuric acid esterification is not as widespread as glucuronidation, the water solubility of the product after sulfuric acid esterification increases and the toxicity decreases, which is more conducive to excreting in vitro. Active sulfating agent 3′-adenosine phosphate-5′phosphoryl sulfuric acid (PAPS), catalyzed by sulfotransferases, provides active sulfate groups to form sulfates from substrates [30].

Flavonoids are also ubiquitous in fruits and vegetables, and people eat a certain amount of fruits and vegetables every day, so flavonoids must be absorbed by the human body. The human body’s absorption of flavonoids in fruits and vegetables affects the biotransformation of drugs and other substances in the human body, and may even lead to various toxic and side effects. Cytochrome P450, monooxygenase and other phase I metabolic enzymes are a group of important enzymes which play an important role in the metabolism of hydrophobic endogenous substrates (sterols, prostaglandins, fatty acids) and the absorption of exogenous substances (drugs, carcinogens, food ingredients, pollutants) [71]. The interaction between flavonoids and cytochrome P450 will produce a series of side effects, affecting the concentration of drugs in the blood, causing excessive or insufficient pharmacological effects, and may even be toxic. Flavonoids may also increase the activity of CYP-carcinogenic substances by inducing the expression of CYPs or the activity of CYPs [72]. However, flavonoids are often used as an agonist of aromatic receptor (AhR) to induce the active expression of CYP1A1 and CYP1A2 (CYP1A1 and CYP1A2 are responsible for the activation of carcinogenic substances such as benzo-α-pyrene, dimethylbenzoanthrene, aflatoxin B1 and heterocyclic aromatic amines). Flavonoids such as galangin, quercetin, diosmin and geraniol can increase CYP1A1 gene expression [73,74].

Flavonoids, especially isoflavones, are structurally similar to estrogen. Excessive or improper intake of flavonoids can lead to imbalance in hormone metabolism and endocrine regulation in humans, resulting in various adverse consequences. H Stopper et al. [75] reviewed the genotoxic effects of phytoestrogens and concluded that genistein, coumarin, quercetin and resveratrol had genotoxic effects in vitro [75]. Therefore, it is necessary to investigate the metabolic pathways of different flavones.

### 3.3. Pharmacokinetic and Prostate Distribution

The aerial part of *G. uralensis* has been studied in our research group for many years, but it has not been used in clinical practice. In our previous study [1], 1 g/kg of the aerial part of *G. uralensis* extract was determined to be the best effective dose to induce anti-CNP activity. According to the conversion ratio between human and rat, the dosage of human should be 0.159 g/kg. Our previous study found that the classification of acute toxicity of ethanol extract from aerial parts of *G. uralensis* belonged to the non-toxic level, with the maximum tolerance dose of 96.128 g/kg, which was equivalent to the daily lethal dose of adults >500 g, and no genetic toxicity was found [50].

The results of the pharmacokinetic studies found that two or even three peaks appeared in the concentration–time curves of the four components after administration, indicating that four flavonoids in vivo may exhibit hepatoenteric circulation and gastric emptying. In the liver, hepatocytes constantly generate bile acids and secrete bile. The cells with phagocytosis are mainly sinus endothelial cells and Kupffer cells. It is possible that flavonoids are absorbed by hepatocytes and secreted into the bile. They are reabsorbed in the intestine, forming a liver-gut cycle, and showing a bimodal phenomenon. However, this reason needs to be studied.

According to the concentration in prostate, kaempferol has the highest concentration. At present, some studies have shown that kaempferol can treat prostate diseases. For example, Wang X [76] analyzed the effect and mechanism of kaempferol on benign prostatic hyperplasia by establishing a rat model of benign prostatic hyperplasia induced by testosterone. It was found that kaempferol can inhibit the effect induced by dihydrotestosterone, continuously reduce the prostate index of rats, improve the pathological characteristics, and show androgen-like activity. It can be used as a selective androgen receptor regulator, contributing to androgen.

## 4. Materials and Methods

### 4.1. Materials and Reagents

Raw materials: The experimental sample of *G. uralensis* Fisch. was collected from Hedong Township, Guazhou County, Jiuquan City, Gansu Province. It was identified as *G. uralensis* Fisch. by Professor Wang Wenquan of The Institute of Medicinal Plant Development. The sample was naturally dried and crushed and then passed through a 40-mesh sieve. The sample was stored until further use.

Water (Wahaha Group Co., Ltd., (Hangzhou, China) No.: 202012013221TJ), ethanol (analytical pure, Beijing Chemical Plant, No.: 20201120), methanol (analytical pure, Beijing Chemical Plant, No.: 210220), methanol (FISHER, (Shanghai, China) chromatography pure, LOT: 206403), acetonitrile (FISHER, chromatography pure, LOT: 210774), formic acid (FISHER, chromatography pure, No: 200720), puerarin (LOT: S02M9B54875), vicenin-2 (LOT: P01F9F54173), isoschaftoside (LOT: Z16A10B95343), schaftoside (LOT: HA0828XA13), rutin (LOT: T27F10Z81699), liquiritin (LOT: Z10J8X39611), isoquercitrin (LOT: P25J9F65872), isoliquiritin (LOT: C09J7G17535), vitexin (LOT: Y26J9H66647), pinocembrin (LOT: T12J9F65510), luteolin (LOT: C24M8Q36543), naringenin (LOT: YJ0603HA13), genistein (LOT: H30A9Z69019), diosmetin (LOT: T22F7X9844), isorhamnetin (LOT: P23A9F68614), kaempferol (LOT:C26J8Y38642) and retrochalcone (LOT: ZS0903BA13) were used in this study. The above standard products were obtained from Shanghai Yuanye Biotechnology Co., LTD (Shanghai, China).

### 4.2. Preparation of the Aerial Part of Glycyrrhiza Uralensis Sample

The aerial part of *G. uralensis* was subjected to 70% ethanol reflux for extraction. The liquid ratio was 8:1. The extraction was performed twice for 90 min, and then it was concentrated and dried into powder for standby. The HPLC chromatogram of the test substance was obtained by the method determined by our group. The concentrations of puerarin, luteolin, kaempferol and pinocembrin in the extracts were calculated to be 3.89, 5.422, 7.281 and 12.698, μg/g respectively. The HPLC chromatogram is shown in Figure 17.

### 4.3. Animals

Ninety-six clean and healthy male Sprague Dawley (SD) rats weighing 180 ± 20 g were used as experimental animals. They were purchased from Beijing Viton Lihua Experimental Animal Technology Co., Ltd. (Tyler, TX, USA) (Animal Certificate No.: SLXD (Beijing)-20200902021). The rats were fed at a temperature ranging from 22 °C to 24 °C under 5% relative humidity. They were comfortably caged for 5 days with free access to water and food to allow the rats to adapt to the surrounding environment. Subsequently, the rats were fasted for 12 h before being administrated the partial extract from the aerial part of *G. uralensis*. Animal feed and facilities were provided by the Laboratory Animal Center of the Institute of Medicinal Plant Development, Peking Union Medical College (Beijing, China). All rats were handled according to the National Guidelines for the Care and Use of Laboratory Animals, and all animal experiments were approved by the Animal Ethics Committee of the Chinese Academy of Medical Sciences and Institute of Medicinal Plant Development (approval no. SLXD-20200902021).

### 4.4. Preparation of Plasma, Urine, Fecal and Prostate Samples

#### 4.4.1. Metabolic Characteristics of Four Representative Flavonoids

Forty-eight rats were randomly divided into four groups, namely plasma (*n* = 6), prostate (*n* = 30, there were five time points, with six individuals for each time point), excretion (*n* = 6) and blank (*n* = 6). The plasma, prostate, and excretion groups were gavaged with 1 g/kg of *Glycyrrhiza uralensis* sample, whereas the blank group was administered 0.5% CMC-Na. Biological samples were collected after the mice were dosed once.

Orbital blood (300 μL) was collected from the plasma group at 0.25, 0.5, 1, 2, 4, 6, 8, 12, 24, 36 and 48 h from postorbital venous plexus veins after administration in a heparitinated plastic centrifuge tube. The tube was placed at room temperature for 1 h; then, it was centrifuged at 4 °C (10,000 rpm, 10 min). Upper plasma of different time points was separated for 50 μL and mixed and shaken well. Mixed plasma was collected at 50 μL and the plasma samples (50 μL) were vortex-mixed with 250 μL of methanol. The solution was centrifuged at 10,000 rpm for 10 min to separate the protein from the organic phase, and then the upper supernatant was collected. Then, the supernatant was transferred and evaporated to dryness under a gentle stream of nitrogen at room temperature. The residue was reconstituted in 50 µL methanol, mixed for 1 min, and centrifuged at 10,000 rpm for 10 min. At last, 20 μL of the resulting supernatant was applied to UPLC–MS/MS analysis. 

Rats in the prostate group were sacrificed with cervical dislocation at 0.5, 1.5, 2.5, 6 and 8 h after administration. The prostate samples (1 g) were homogenized in an equal volume of normal saline. Then, the mixed prostate homogenate supernatant was treated in the same manner as the plasma.

Urine and feces samples were collected at 0–4, 4–8, 8–12, 12–24, 24–36 and 36–48 h after administration in the excreta group. The supernatant (100 μL) was centrifuged and added to the activated SPE column. After washing with 1 mL water, the new test tube was replaced, followed by elution with 1 mL methanol. The supernatant was centrifuged and injected for detection. The feces were dried, weighed, and ground into a powder. The powder (0.05 g) was added to a 5 mL volumetric flask to which methanol was added for calibration. The powder was ultrasonically dissolved for 15 min after low-speed centrifugation, and 100 μL of the fecal supernatant samples was added to the activated SPE column. The follow-up procedure was the same as that for the urine pretreatment, and the blank urine and fecal samples were used in this process.

#### 4.4.2. Content Determination of Four Representative Flavonoids

Forty-eight rats were randomly divided into four groups, namely plasma (*n* = 6), prostate (*n* = 30, there were five time points, with six individuals for each time point), and blank (*n* = 6). 

Blood samples of about 300 μL were collected into heparinized centrifuge tubes from the fossa orbitalis vein at 3 h after administration on days 1, 3, 5, 9, 11, 13, 15, 17, 19, 21, 25, 27, 29, 31 and at 0.25, 0.5, 1, 2, 4, 6, 8, 12, 24, 36 and 48 h on day 31 in six healthy rats after oral administration of the aerial part of *G. uralensis* extract (1 g/kg) every day. For clean-up, the plasma samples (50 μL) were vortex-mixed with 10 μL of IS and 250 μL of methanol. The solution was centrifuged at 10,000 rpm for 10 min to separate the protein from the organic phase, and then the upper supernatant was collected. Then, the supernatant was transferred and evaporated to dryness under a gentle stream of nitrogen at room temperature. The residue was reconstituted in 50 µL methanol, mixed for 1 min, and centrifuged at 10,000 rpm for 10 min. At last, 5 μL of the resulting supernatant was applied to HPLC–MS/MS analysis. 

Rats in the prostate group were sacrificed with cervical dislocation at 0.5, 1.5, 2.5, 6 and 8 h after administration on day 31. The prostate samples (1 g) were homogenized in an equal volume of normal saline. Then, the prostate homogenate supernatant was treated in the same manner as the plasma.

### 4.5. Preparation of Calibration Standards and Quality Control (QC) Samples

Plasma calibration curves were prepared by adding the standard and IS solution into the control plasma to obtain the final concentrations for four flavones (0.1, 0.2, 0.5, 1, 2, 5, 10, 20, 50, 100, 200, 500, 1000 ng/mL). Plasma quality control (QC) samples were prepared in the same manner at low, medium and high concentrations (puerarin, 0.5, 10, 50 ng/mL; luteolin, 0.2, 2, 5 ng/mL; kaempferol, 0.1, 0.5, 2 ng/mL; pinocembrin, 0.1, 0.5, 2 ng/mL), respectively.

Calibration standards for the prostate were established by adding standard and IS solution into the control tissue supernatant to obtain the final concentrations for four flavones (0.1, 0.2, 0.5, 1, 2, 5, 10, 20, 50, 100, 200, 500, 1000 ng/mL). Furthermore, prostate QC samples were prepared in the same manner by spiking into the prostate tissue homogenates at the appropriate volumes of different stock solutions to achieve the low, medium and high concentrations (puerarin, 0.1, 1, 5 ng/mL; luteolin, 0.2, 5, 10 ng/mL; kaempferol, 2, 100, 200 ng/mL; pinocembrin, 2, 50, 100 ng/mL), respectively. The IS (nimodipine) concentration was 2 µg/mL in all calibration standards and QCs. All of the solutions were stored at 4℃ until analysis.

### 4.6. Instrumentation and Analytical Conditions

#### 4.6.1. Metabolic Characteristics of Four Representative Flavonoids

UPLC-Q-Exactive Orbitrap-MS analysis was performed on a Dionex Ultimate 3000 ultra-high performance liquid system (ThermoFisher) consisting of a binary pump, a DAD detector, a column chamber, an automatic sampler and an online degasser. The sample was separated on a Waters ACQUITY UPLC HSS T3 C18 column (2.1 mm × 100 mm, 1.8 μm), with a column temperature of 30 °C, automatic sampler temperature of 10 °C, and flow rate of 0.2 mL·min^−1^. The mobile phases A and B were acetonitrile with 0.1% formic acid and aqueous solution with 0.1% formic acid, respectively. The gradient elution profile was as follows: 0–10 min, 100% B; 10–20 min, 100–70% B; 20–25 min, 70–60% B; 25–30 min, 60–50% B; 30–40 min, 50–30% B; 40–45 min, 30–0% B; 45–60 min, 0% B; 60–60.1 min, 0–100% B; 60.1–70 min, 100%. The injection volume of the sample was 5 μL.

The MS conditions were as follows: detection mode: positive and negative ion modes; scan range: 100–1200; resolution: 70,000; sheath gas flow rate: 40; aux gas flow rate: 15; spray voltage [kV]: 3.20; capillary temp: 320 °C; aux gas heater temp: 350 °C; AGC target: 3,000,000; maximum IT (ms): 100; sheath gas: 40/45; aux gas flow: 11/10; sweep gas: 0; S-lens RF level: 55; and aux gas heater temperature: 220 °C. 

Feature peak extraction in raw files was performed using Compound discover 3.2 software and was identified simultaneously using Mzcloud and the local TCM chemical composition database, with the screening criteria of Mzvault best match score >70 points.

#### 4.6.2. Content Determination of Four Representative Flavonoids

The analysis was performed on the AB SCIEX QTRAP 5500 (USA) triple quadrupole mass spectrometer, which was performed in SRM and positive and negative ionization modes. LC separation is run on XSelect HSS T3 analytical column (2.1 × 100 mm, 2.5 μm, Waters), equipped with Exion LC AD system (AB Sciex) infinite binary pump. Acetonitrile was used as solvent A and 0.1% formic acid aqueous solution was used as solvent B and the flow rate was 0.1 mL/min. Using gradient elution, the initial condition is that 90% solvent B drops to 0% B within 17 min and maintains it for 2 min, returns to the initial 100% B at 19.1 min, and ends an analysis at 22 min. The temperature of the column oven is maintained at 35 °C. The sample is kept at 10 °C and the injection volume is 5 μL. The MS parameters are as follows: ESI ion source temperature is 500 °C; curtain gas: 30 psi; collision-activated dissociation (CAD) gas settings: medium, ion spray voltage: −4500 V, ion gas 1 and 2: 50 psi. The data acquisition software is AB SCIEX Analyst 1.7.1. The specific parameters are as Table 8.

### 4.7. Method Validation on Pharmacokinetics and Prostate Distribution

Method validation for this assay was performed according to the guiding principles for the quantitative analysis of biological samples (Chinese Pharmacopoeia, 2020 edition).

#### 4.7.1. Selectivity

The selectivity of the method was investigated by comparing the chromatograms of the rat plasma control (plasma/prostate homogenates), the homogenates of the rat plasma/prostate control spiked with four analytes and IS (nimodipine), the homogenates of the rat plasma/prostate samples obtained 4 h after the oral administration of aqueous extract of the aerial part of *G. uralensis*, as well as the mixture of standard and IS solution, respectively. The homogenates of the rat plasma/prostate control were obtained from three random rats that were used in the pharmacokinetics study. All the samples were prepared in the same way as in the case of the plasma and tissue sample pretreatment.

#### 4.7.2. Calibration Curves 

Calibration curves were determined by plotting the peak area ratio of each analyte to the IS (Y-axis) against the nominal value of the analyte (X-axis), and were constructed with a weighting (1/X) factor by the least-squares regression analysis.

#### 4.7.3. Extraction Recovery and Matrix Effect

Extraction recovery and the matrix effect of IS and analytes at three QC concentrations (low, medium and high) were assayed in sets of three replicates. The recovery values of the extraction of four analytes were evaluated by comparing the peak areas from the plasma samples/prostate homogenates spiked before protein precipitation to those from the samples spiked after the pretreatment. The matrix effects were determined by comparing the peak areas of analytes spiked to extract the control sample with those from the neat standard solution at the same concentration.

#### 4.7.4. Accuracy and Precision

The intra-day precision and accuracy were evaluated by analyzing the QC samples at different concentrations three times in a single day. The inter-day precision and accuracy were assessed by analyzing three replicates at each concentration level on three consecutive days. The accuracy was calculated as the relative error (RE), and the precision was calculated as the relative standard deviation (RSD).

#### 4.7.5. Stability

Stability investigation of the four components in the plasma and prostate homogenates was performed by analyzing replicates (*n* = 3) of the three levels of QC samples under three different conditions. Room temperature stability was assessed after the exposure of untreated QC samples at room temperature for 24 h. Freeze and thaw stability was evaluated after three freeze (−20 °C)–thaw (room temperature) cycles.

## 5. Conclusions

In this study, UPLC–MS/MS was used to analyze the chemical constituents of the biological samples of rats perfused with aerial parts of *G. uralensis*. The number of prototype components detected in the prostate, plasma, urine and feces samples was 22, 17, 22 and 11, respectively. A comprehensive analysis indicated that the majority of prototypes and metabolites were detected in the urine samples; thus, we speculate that flavonoids in the aboveground part of *G. uralensis* are mainly excreted through urine.

To explore the metabolic pathways of four representative bioactive flavonoids (puerarin, luteolin, kaempferol and pinocembrin) in rats, we studied the metabolism of four analytes from *G. uralensis* after oral administration of the aerial parts of *G. uralensis*. In order to explore the metabolic pathway of flavonoids in rats, we studied the metabolism of four representative bioactive flavonoids (1–4) in the aerial parts of *Glycyrrhiza uralensis* Fisch. after oral administration in rats. Finally, four prototype components were detected in plasma, prostate, urine and feces, and the metabolic pathway of kaempferol in feces was discussed for the first time. The results showed that kaempferol was mainly excreted in feces in the form of prototype, quercetin, sulfated, methylated and glucuronic acid conjugate. Two unreported metabolites of phase II—3-M1 and 3-M2, which are methylated and sulfated products of luteolin, were detected in plasma, urine and feces. The main metabolic pathways of flavonoids in aerial parts of *Glycyrrhiza uralensis* Fisch are oxidation reaction in phase I metabolism and methylation, sulfation and glucuronidation reaction in phase II metabolism.

In this study, we first investigated the metabolism of puerarin, kaempferol, luteolin and pinocembrin in the rat prostate, and found that the metabolites of puerin in the prostate are mainly hydrolysis and glucose-conjugated products, the metabolites of kaempferol and luteolin are mainly methylation and glucuronidation, and the metabolites of pinocembrin are mainly naringenin, oxidation, sulfation, methylation and glucuronidation products. Analysis of metabolites in the prostate revealed that methylation and glucuronidation metabolites were the main metabolites detected in the prostate, which might be a potential component for the treatment of prostate diseases and may provide a basis for further research.

This paper described a simple, sensitive and validated HPLC–MS/MS method for simultaneous determination of puerarin, luteolin, kaempferol and pinocembrin in rat plasma and prostate after the oral administration of the aerial parts of *G. uralensis*, and investigated their pharmacokinetic and prostate distribution studies as well. This method was sensitive, with high accuracy and met all requirements in the bioanalytical method. It was successfully applied to the pharmacokinetic and prostate distribution studies of four active components of the aerial parts of *G. uralensis*. This study may shed new light on the biological behavior of four analytes in vivo and help explain relevant pharmacological actions against chronic non-bacterial prostatitis related to the aerial parts of *G. uralensis*.

## Figures and Tables

**Figure 1 molecules-27-03245-f001:**
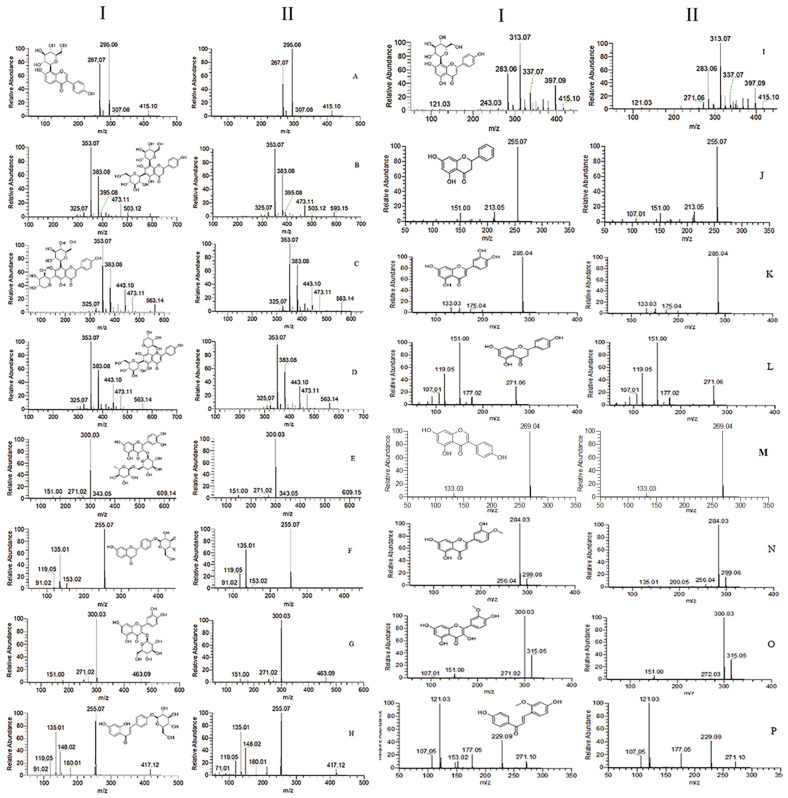
Spectra of 16 components in reference solution (I) and sample (II). (**A**) Puerarin; (**B**) vicenin-2; (**C**) isoschaftoside; (**D**) schaftoside; (**E**) rutin; (**F**) liquiritin; (**G**) isoquercitrin; (**H**) isoliquiritin; (**I**) vitexin; (**J**) pinocembrin; (**K**) luteolin; (**L**) naringenin; (**M**) genistein; (**N**) diosmetin; (**O**) isorhamnetin; (**P**) retrochalcone.

**Figure 2 molecules-27-03245-f002:**
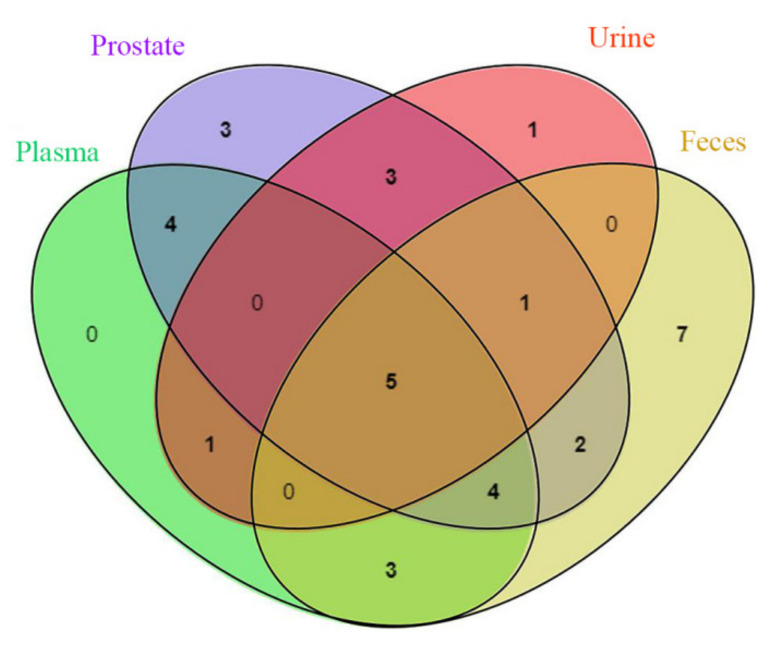
Wayne diagram of the number of prototype components in plasma, prostate, urine and feces.

**Figure 3 molecules-27-03245-f003:**
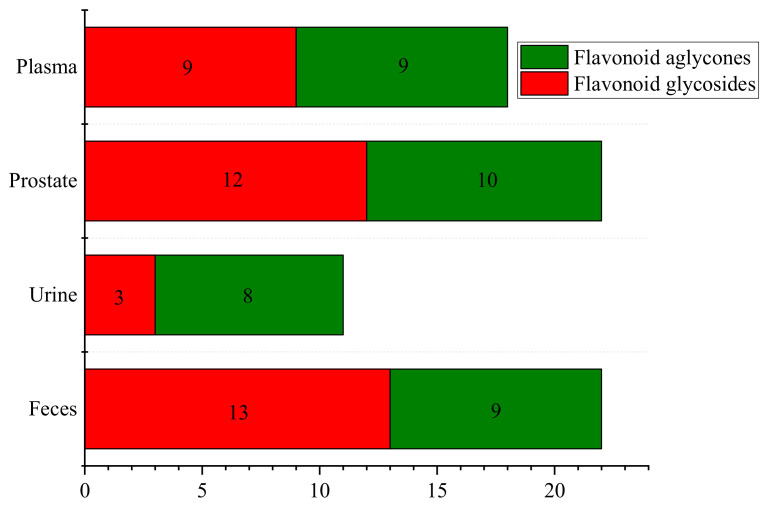
Strip graph of flavonoid glycosides and flavonoid aglycones in plasma, prostate, urine and feces.

**Figure 4 molecules-27-03245-f004:**
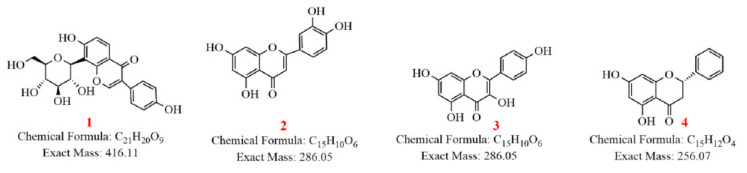
Structure of four representative flavonoids from aerial parts of *Glycyrrhiza uralensis*; 1—puerarin (isoflavones); 2—luteolin (flavonoids); 3—kaempferol (flavonol); 4—pinocembrin (flavone).

**Figure 5 molecules-27-03245-f005:**
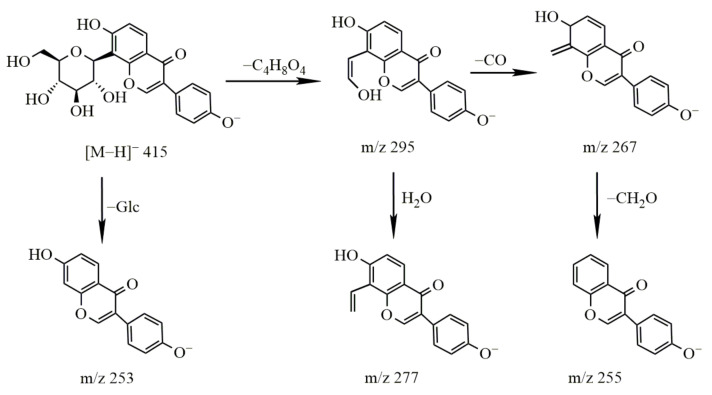
The cleavage pathway of puerarin.

**Figure 6 molecules-27-03245-f006:**
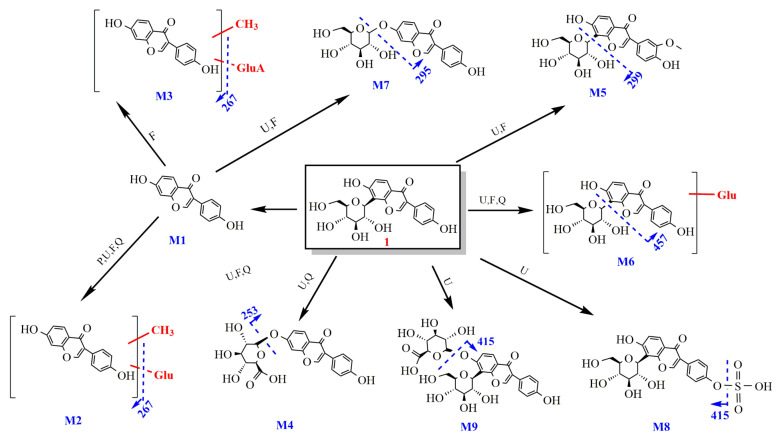
The metabolic pathway of puerarin.

**Figure 7 molecules-27-03245-f007:**
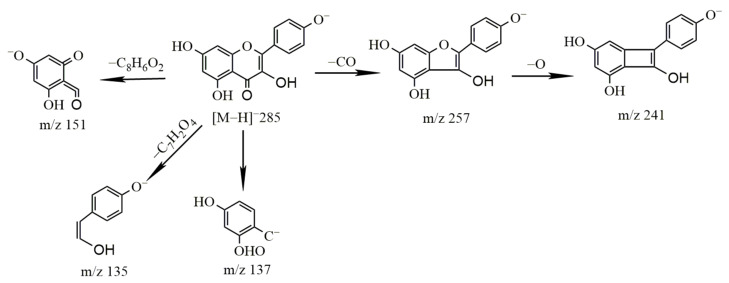
The cleavage pathway of kaempferol.

**Figure 8 molecules-27-03245-f008:**
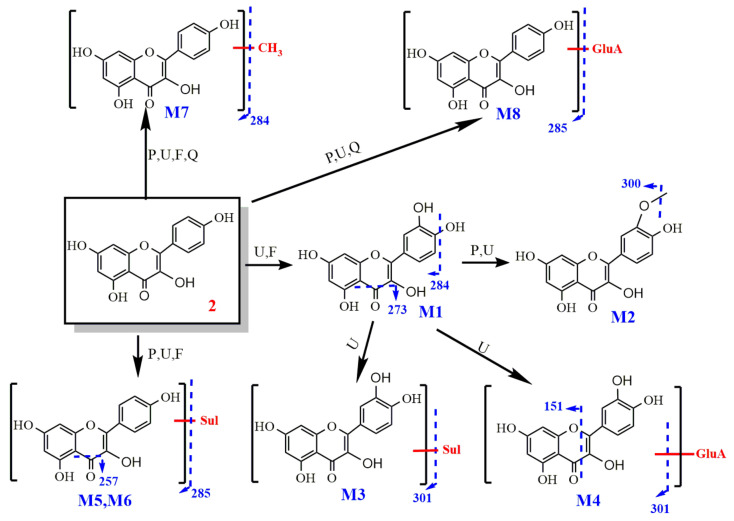
The metabolic pathway of kaempferol.

**Figure 9 molecules-27-03245-f009:**
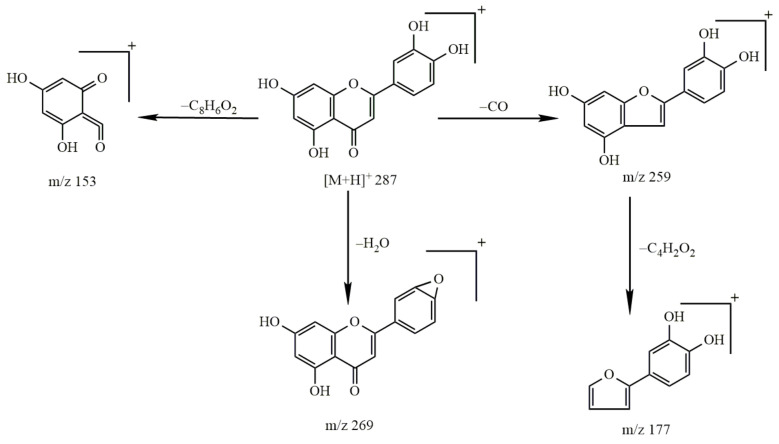
The cleavage pathway of luteolin.

**Figure 10 molecules-27-03245-f010:**
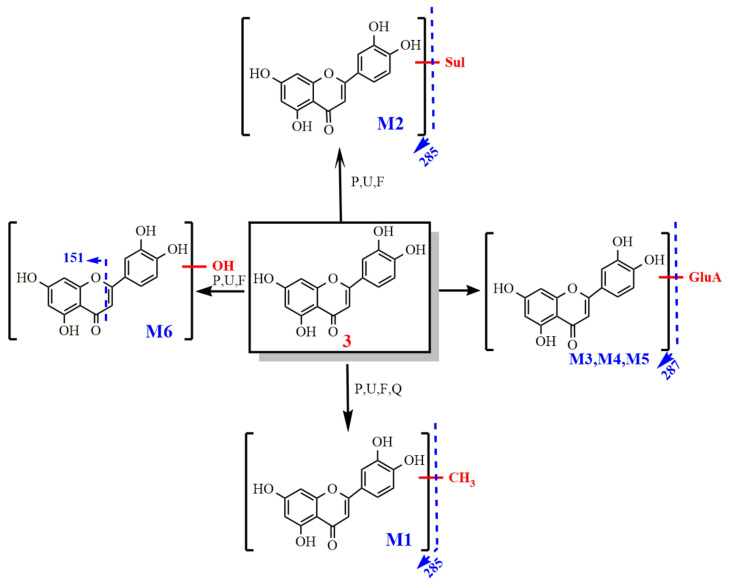
The metabolic pathway of luteolin.

**Figure 11 molecules-27-03245-f011:**
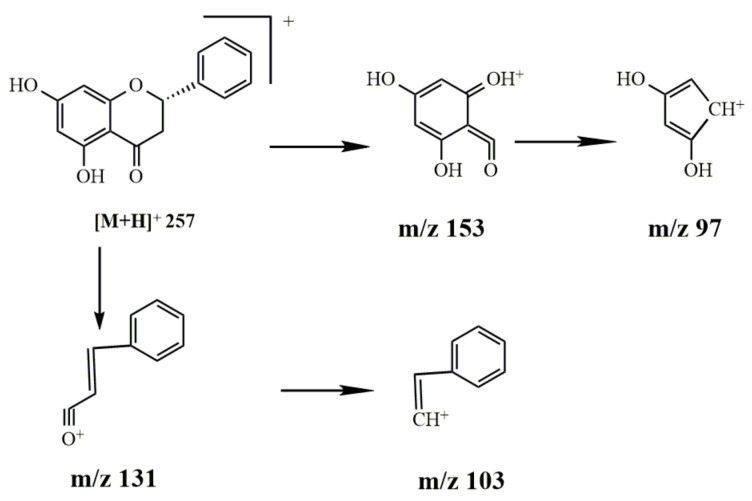
The cleavage pathway of pinocembrin.

**Figure 12 molecules-27-03245-f012:**
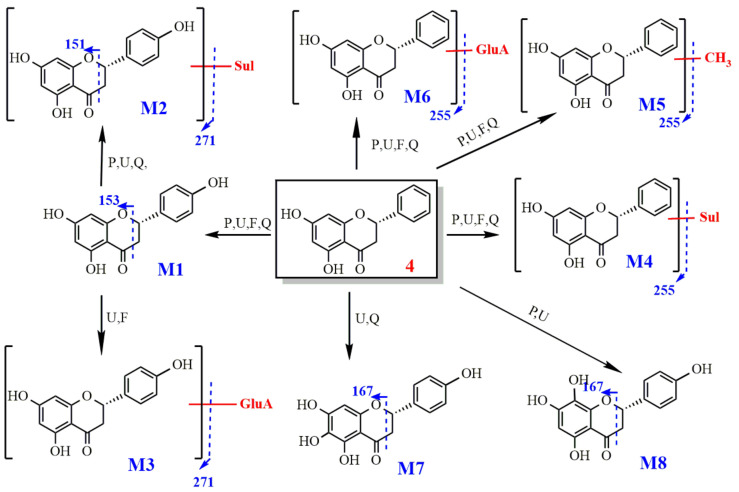
The metabolic pathway of pinocembrin.

**Figure 13 molecules-27-03245-f013:**
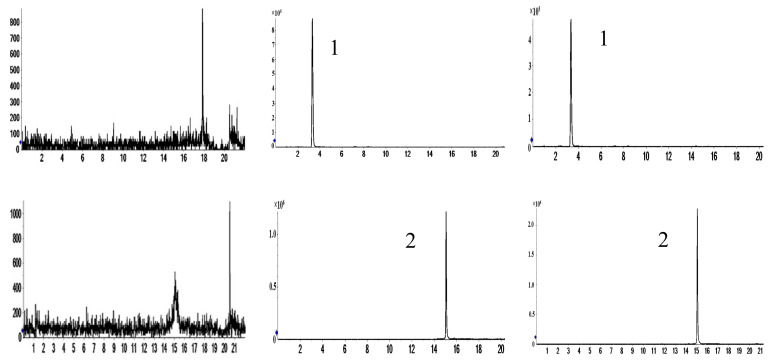
Representative MRM chromatograms of analytes in different samples. (**a**) Blank plasma; (**b**) blank plasma + reference substance+IS; (**c**) plasma samples after oral administration of the aerial parts of *G. uralensis* for 4 h. (1) Puerarin; (2) luteolin; (3) kaempferol; (4) pinocembrin; (5) IS (nimodipine).

**Figure 14 molecules-27-03245-f014:**
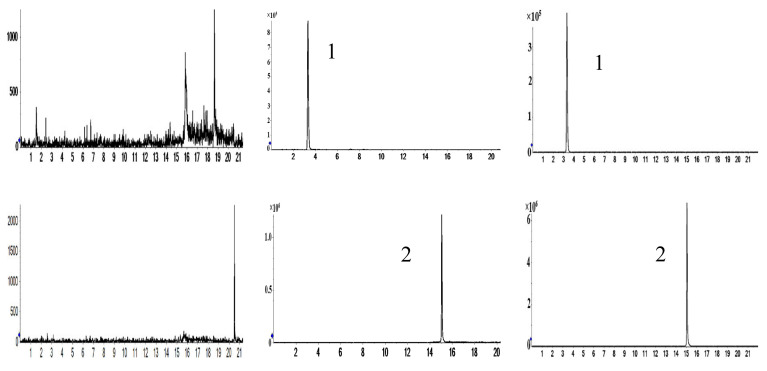
Representative MRM chromatograms of analytes in different samples. (**a**) Blank prostate; (**b**) blank prostate + reference substance+IS; (**c**) prostate samples after oral administration of the aerial parts of *G. uralensis* for 4 h; (1) puerarin; (2) luteolin; (3) kaempferol; (4) pinocembrin; (5) IS (nimodipine).

**Figure 15 molecules-27-03245-f015:**
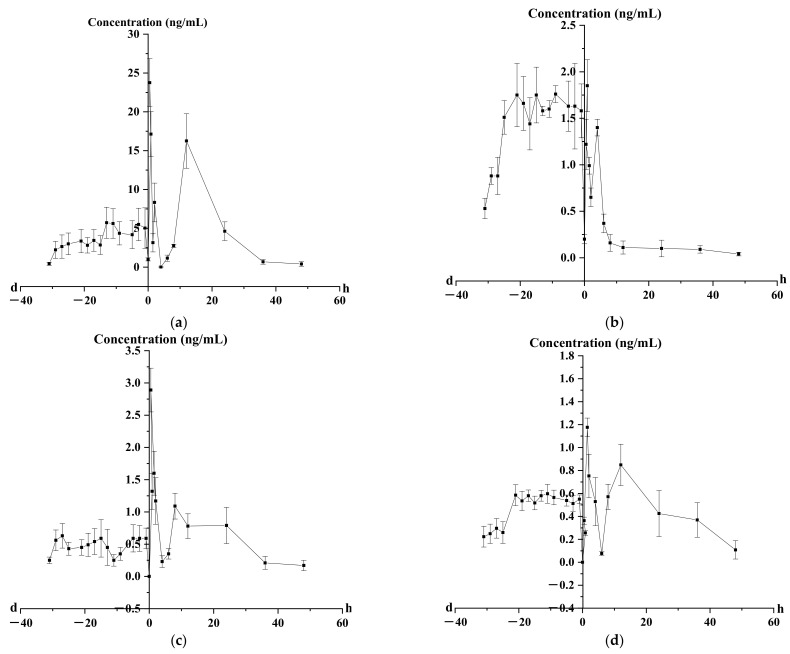
Blood concentration–time curve of four analytes after oral administration of the aerial parts of *G. uralensis.* (**a**) Puerarin; (**b**) luteolin; (**c**) kaempferol; (**d**) pinocembrin (d—day; h—hour; the time before administration on day 31 was taken as the y-axis position, before the y-axis indicates the concentration at different days, and after the y-axis represents the concentration at different time points after the last day of administration).

**Figure 16 molecules-27-03245-f016:**
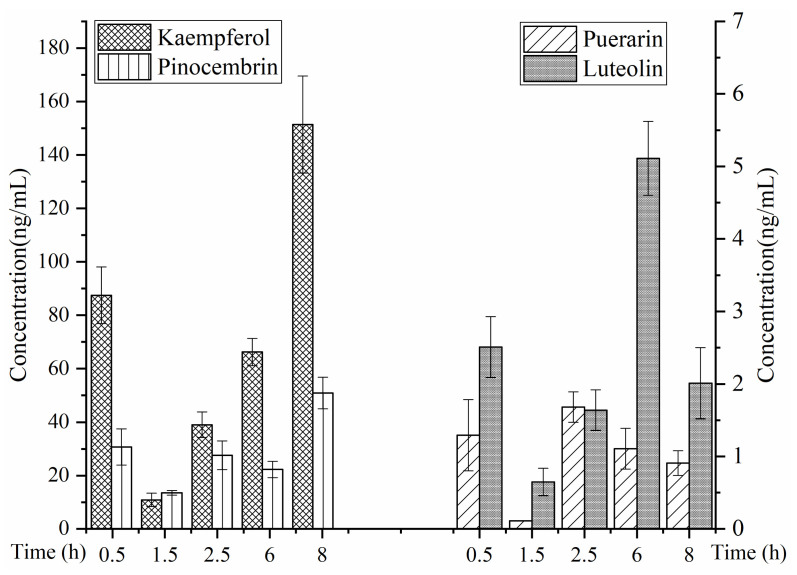
Prostate tissue distribution of four analytes after oral administration of the aerial parts of *G. uralensis* (*n* = 6).

**Figure 17 molecules-27-03245-f017:**
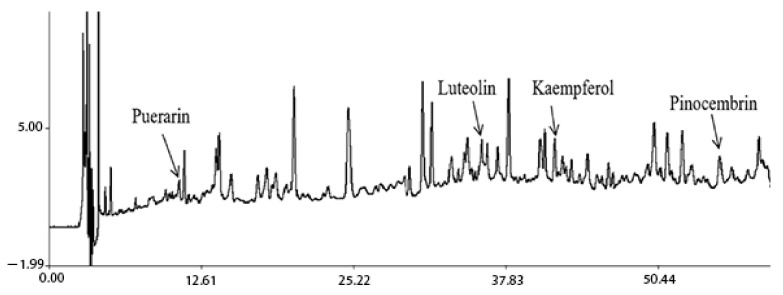
HPLC chromatogram of extracts from aerial parts of *Glycyrrhiza uralensis*.

**Table 1 molecules-27-03245-t001:** Distribution of protoflavones in biological samples.

No.	RT [min]	Name	Formula	Predicted	Measured	HR-MS	MS^2^	∆ (ppm)	Plasma	Prostate	Urine	Feces
**1**	33.922	8-Prenylnaringenin [9]	C_20_H_20_O_5_	339.1227	339.1231	[M − H]^−^	219, 133, 119	−0.44				+
**2**	22.566	Astragalin [7]	C_21_H_20_O_11_	447.0922	447.0925	[M − H]^−^	285, 284, 255, 227, 151	0.21	+	+		+
**3**	23.924	Baicalin [7]	C_15_H_10_O_5_	269.0445	269.0451	[M − H]^−^	241, 223, 197, 169, 136	1.14		+		+
**4**	25.602	Calycosin [10]	C_16_H_12_O_5_	285.0758	285.0764	[M + H]^+^	270, 253, 225, 137	−0.78		+		
**5**	27.92	Diosmetin *	C_16_H_12_O_6_	299.0550	299.0555	[M − H]^−^	284, 256, 227, 180, 151	−0.63	+	+	+	+
**6**	22.633	Diosmin [11]	C_28_H_32_O_15_	609.1814	609.1833	[M + H]^+^	463, 301, 286, 258, 85	0.85		+		+
**7**	27.498	Genistein *	C_15_H_10_O_5_	269.0451	269.0444	[M − H]^−^	225, 201, 181, 151, 133	−0.49	+	+		+
**8**	22.733	Hesperidin [11]	C_28_H_34_O_15_	609.1814	609.1818	[M − H]^−^	325, 301, 286, 242, 164, 151, 125	0.6				+
**9**	21.784	Hyperoside [12]	C_21_H_20_O_12_	463.0871	463.0894	[M − H]^−^	300, 271, 255, 243, 151	0.52	+	+		
**10**	21.476	Isoliquiritigenin *	C_15_H_12_O_4_	257.0808	257.0812	[M + H]^+^	242, 211, 147, 137, 119, 91, 81	−1.03		+	+	
**11**	36.151	Isoliquiritin *	C_21_H_22_O_9_	417.1180	417.1183	[M − H]^−^	255, 180, 148, 135, 119, 108, 91	0.16		+		+
**12**	23.188	Isoquercitrin *	C_21_H_20_O_12_	463.0871	463.0875	[M − H]^−^	300, 271, 179, 151	2.62	+	+		+
**13**	21.683	Isorhamnetin *	C_16_H_12_O_7_	315.0499	315.0505	[M − H]^−^	300, 283, 255, 151, 107	0.15	+		+	
**14**	22.252	Isorhamnetin-3-O-nehesperidine [13]	C_28_H_32_O_16_	625.1763	625.1777	[M + H]^+^	317, 302, 85, 71	0.35				+
**15**	22.257	Isorhamnetin-3-O-rutinoside [13]	C_28_H_32_O_16_	623.1607	623.1608	[M − H]^−^	417, 314, 299, 255	0.27		+		
**16**	20.61	Isoschaftoside *	C_26_H_28_O_14_	563.1395	563.1396	[M − H]^−^	473, 443, 413, 383, 353, 325, 191	0.21		+		
**17**	18.738	Isovitexin [14]	C_21_H_20_O_10_	433.1129	433.1138	[M + H]^+^	415, 397, 379, 367, 337, 313, 283	0.12		+	+	
**18**	22.557	Kaempferol [14]	C_15_H_10_O_6_	287.0550	287.0556	[M + H]^+^	258, 165, 153, 121	−0.14	+	+	+	+
**19**	21.864	Kaempferol3-glucorhamnoside [14]	C_27_H_30_O_15_	593.1501	593.1505	[M − H]^−^	285, 284, 255, 229, 227, 151	0.05				+
**20**	45.936	Liquiritin *	C_21_H_22_O_9_	417.1180	417.1185	[M − H]^−^	255, 153, 135, 119, 91	0.3	+			+
**21**	3.812	Luteolin *	C_15_H_10_O_6_	285.0394	285.0399	[M − H]^−^	257, 241, 199, 151	−0.4	+	+	+	+
**22**	29.342	Naringenin *	C_15_H_12_O_5_	271.0601	271.0607	[M − H]^−^	177, 165, 151, 119, 107, 93, 83, 65	0.08	+	+		
**23**	22.511	Naringin [15,16]	C_27_H_32_O_14_	579.1708	579.1711	[M − H]^−^	313, 271, 151, 119, 107	−0.28				+
**24**	33.203	Ononin [17]	C_22_H_22_O_9_	431.1337	431.1346	[M + H]^+^	269, 254, 237, 213, 118, 107	−0.37	+			+
**25**	20.822	Orientin [17]	C_21_H_20_O_11_	447.0922	447.0926	[M − H]^−^	357, 327, 297, 285, 269, 217, 151	0.18				+
**26**	21.68	Pinocembrin *	C_15_H_12_O_4_	255.0652	255.0657	[M − H]^−^	213, 151, 107, 83	−1.08	+	+	+	+
**27**	19.604	Puerarin *	C_21_H_20_O_9_	415.1024	415.1027	[M − H]^−^	361, 307, 295, 277, 267, 253, 109	−0.03	+	+	+	
**28**	21.779	Quercetin [18]	C_15_H_10_O_7_	301.0343	301.0347	[M − H]^−^	273, 178, 151, 121	−1.02		+	+	+
**29**	27.347	Retrochalcone *	C_16_H_14_O_4_	271.0965	271.0967	[M + H]^+^	229, 177, 121, 107	−0.8	+	+	+	+
**30**	21.345	Rutin *	C_27_H_30_O_16_	609.1450	609.1452	[M − H]^−^	300, 271, 255, 244, 178, 151	0.07	+	+		
**31**	20.398	Schaftoside *	C_26_H_28_O_14_	563.1395	563.1396	[M − H]^−^	473, 443, 413, 383, 353, 325, 191	−0.12	+			+
**32**	19.745	Vicenin-2 *	C_27_H_30_O_15_	593.1501	593.1502	[M − H]^−^	503, 473, 383, 353, 325, 297	0.11			+	
**33**	17.75	Vitexin *	C_21_H_20_O_10_	433.1129	433.1128	[M + H]^+^	415, 397, 337, 323, 313, 283, 121	0.08	+			
**34**	25.965	Wogonin [19]	C_16_H_12_O_5_	285.0758	285.0764	[M − H]^−^	270, 253, 177, 150	−0.77	+	+		+

* Identified by comparing them with reference standards.

**Table 2 molecules-27-03245-t002:** Distribution of four representative flavonoid metabolites in rats.

No.	Formula	RT [min]	Δ (ppm)	Calc. MW	HR-MS	MS	Identification	Plasma	Urine	Feces	Prostate
**1 ***	C_21_H_20_O_9_	19.592	0.53	416.11095	[M − H]^−^	295, 277, 267, 253	Puerarin	+	+	+	+
**1-M1**	C_15_H_10_O_4_	24.857	0.18	254.238	[M + H]^+^	145, 137	Daidzein				
**1-M2** [23]	C_21_H_20_O_10_	21.683	−0.28	432.10552	[M + H]^+^	415, 313, 283, 267	D-Me-Glu	+	+	+	+
**1-M3** [23]	C_22_H_20_O_10_	24.136	−0.19	444.10556	[M − H]^−^	267, 253	D-Me-GluA			+	
**1-M4** [23]	C_21_H_18_O_10_	20.618	−0.15	430.08993	[M − H]^−^	253	Daidzein-7-O-glucuronid		+		+
**1-M5** [24]	C_22_H_22_O_10_	19.913	0.26	446.12141	[M + H]^+^	327, 299	3′-Methoxy Puerarin		+	+	
**1-M6** [24]	C_27_H_30_O_14_	21.253	0.06	578.16359	[M − H]^−^	457	Puerarin-Glu		+	+	+
**1-M7** [25]	C_21_H_20_O_9_	19.673	0.31	416.11068	[M − H]^−^	295	Daidzin		+	+	
**1-M8** [25]	C_21_H_20_O_12_S	23.896	0.19	496.06764	[M − H]^−^	415	puerarin 4′-O-β-sulfate		+		
**1-M9** [25]	C_27_H_28_O_15_	19.177	0.72	592.14325	[M − H]^−^	415, 295, 253	puerarin-7-O-glucuronide		+		
**2 ***	C_15_H_10_O_6_	24.739	−0.52	286.04759	[M − H]^−^	257, 241, 151, 137, 135	Kaempferol		+	+	+
**2-M1** [26]	C_15_H_10_O_7_	25.616	−0.72	302.04244	[M + H]^+^	284, 273	Quercetin		+	+	
**2-M2** [27]	C_16_H_12_O_7_	27.27	0.2	316.262	[M + H]^+^	300	Isorhamnetin	+	+		
**2-M3** [28]	C_15_H_10_O_10_S	24.754	0.15	381.99953	[M − H]^−^	301, 284, 151	Q-S		+		
**2-M4** [28]	C_21_H_18_O_13_	21.710	1.37	478.07539	[M − H]^−^	301, 284, 151	Q-GluA		+		
**2-M5** [29]	C_15_H_10_O_9_S	24.349	−1.23	366.0041	[M − H]^−^	285, 257, 151, 137	K-S	+	+	+	
**2-M6** [29]	C_15_H_10_O_9_S	24.643	−0.14	366.0045	[M − H]^−^	285, 257, 151, 137	K-S	+	+	+	
**2-M7** [30]	C_16_H_12_O_6_	27.960	−1.28	300.06301	[M − H]^−^	284, 151	K-Me	+	+	+	+
**2-M8** [29]	C_21_H_18_O_12_	22.864	−0.37	462.07966	[M − H]^−^	285, 268, 240	K-GluA	+	+	+	+
**3 ***	C_15_H_10_O_6_	27.34	−0.72	286.04753	[M + H]^+^	269, 259, 177, 153	Luteolin	+	+	+	+
**3-M1** [31]	C_16_H_12_O_6_	29.310	−0.62	300.0632	[M − H]^−^	285, 256	L-Me	+	+	+	+
**3-M2** [31]	C_15_H_10_O_9_S	24.599	−0.16	366.00449	[M − H]^−^	285	L-S	+	+	+	
**3-M3** [31]	C_21_H_18_O_12_	24.919	1.14	462.07964	[M + H]^+^	287	L-7-GluA		+		+
**3-M4** [31]	C_21_H_18_O_12_	22.864	−0.81	462.07945	[M + H]^+^	287	L-4′-GluA	+	+		+
**3-M5** [31]	C_21_H_18_O_12_	22.861	1.44	462.08049	[M + H]^+^	287	L-3′-GluA		+	+	
**3-M6**	C_15_H_10_O_7_	25.656	−0.66	302.04245	[M − H]^−^	273, 151	L+O	+	+	+	
**4 ***	C_15_H_12_O_4_	21.68	−1.08	256.07328	[M + H]^+^	153, 131, 103, 97	Pinocembrin	+	+	+	+
**4-M1** [32]	C_15_H_12_O_5_	33.179	−0.34	272.06838	[M − H]^−^	151, 119	Naringenin	+	+	+	+
**4-M2**	C_15_H_12_O_8_S	27.385	−0.28	352.02519	[M − H]^−^	271, 151, 119	N+S	+	+		+
**4-M3**	C_21_H_20_O_11_	22.978	0.08	448.1006	[M − H]^−^	271, 135, 115	N+GluA		+	+	
**4-M4**	C_15_H_12_O_7_S	23.124	0.38	336.0305	[M − H]^−^	255, 135, 119	P+S	+	+	+	+
**4-M5**	C_16_H_14_O_4_	25.508	−0.53	270.08907	[M − H]^−^	254	P+Me	+	+	+	+
**4-M6** [32]	C_21_H_20_O_10_	21.630	−0.46	432.10545	[M − H]^−^	255, 135, 119	P+GluA	+	+	+	+
**4-M7** [32]	C_15_H_12_O_7_	23.172	2.9	306.07484	[M − H]^−^	167	5,6,7-Trihydroxyflavanone		+		+
**4-M8** [32]	C_15_H_12_O_7_	25.459	−0.37	306.07384	[M − H]^−^	167	5,7,8-Trihydroxyflavanone	+	+		

* Identified by comparing with reference standards; D—daidzein; Q—quercetin; K—kaempferol; L—luteolin; N—naringenin; P—pinocembrin; Me—methylation; Glu—glycosylation; S—sulfating; GluA—glucuronidation.

**Table 3 molecules-27-03245-t003:** Standard curve equations of four analytes in plasma and prostate (*n* = 3).

Sample	Components	Test Range (ng·mL^−1^/ng·g^−1^)	Regression Equation	*r^2^*	Lower Limit of Quantification (ng·mL^−1^/ng·g^−1^)
Plasma	Puerarin	0.5–100	y = 0.006x + 0.0121	0.9994	0.5
	Kaempferol	0.2–5	y = 0.0012x + 0.0003	0.9981	0.2
	Luteolin	0.2–10	y = 0.1701x − 0.0248	0.9994	0.2
	Pinocembrin	0.1–2	y = 0.0137x + 0.001	1	0.1
Prostate	Puerarin	0.1–5	y = 0.0151x + 0.0028	0.9996	0.1
	Kaempferol	1–200	y = 0.0003x + 0.0004	0.9999	1
	Luteolin	0.2–20	y = 0.0072x + 0.0021	0.9998	0.2
	Pinocembrin	0.2–100	y = 0.0026x + 0.0035	0.9984	0.2

**Table 4 molecules-27-03245-t004:** Extraction recovery and matrix effect of four analytes in plasma and prostate (*n* = 3).

Sample	Component	Concentration (μg·mL^−1^/μg·g^−1^)	Matrix Effect ($\bar{x}$ ± *s*, %)	Precision (%, RSD)	($\bar{x}$ ± *s*, %)	Precision (%, RSD)
Plasma	Puerarin	0.5	102.703 ± 3.096	3.015	105.855 ± 9.174	8.666
		10	93.878 ± 2.903	3.092	95.308 ± 6.054	6.352
		50	94.895 ± 1.737	1.831	99.462 ± 1.576	1.584
	Luteolin	0.2	105.091 ± 1.373	1.306	96.367 ± 7.273	7.547
		2	94.444 ± 0.275	0.291	109.244 ± 0.77	0.705
		5	96.414 ± 8.307	8.616	107.082 ± 2.178	2.034
	Kaempferol	0.1	103.623 ± 3.494	3.372	100.6999 ± 3.782	3.756
		0.5	109.066 ± 4.723	4.330	99 ± 2.218	2.304
		2	88.801 ± 3.221	3.627	102.664 ± 2.461	2.397
	Pinocembrin	0.1	82.308 ± 1.538	2.128	99.291 ± 5.258	5.285
		0.5	101.507 ± 1.087	1.071	99.865 ± 1.039	1.040
		2	95.215 ± 1.031	1.082	106.759 ± 2.344	2.196
Prostate	Puerarin	0.100 1.000 5.000	92.691 ± 3.453 90.890 ± 5.368 96.419 ± 1.076	3.725 5.906 1.116	98.925 ± 2.845 104.914 ± 6.361 99.325 ± 1.116	2.876 6.063 1.123
	Luteolin	0.200 5.000 10.000	98.342 ± 5.458 101.592 ± 1.914 100.336 ± 0.302	5.550 1.884 0.301	101.686 ± 4.141 96.951 ± 1.258 98.963 ± 0.380	4.072 1.298 0.384
	Kaempferol	2.000 100.000 200.000	99.660 ± 1.640 98.652 ± 0.969 105.671 ± 1.823	1.646 0.983 1.725	104.266 ± 3.086 96.681 ± 0.538 96.512 ± 1.562	2.960 0.556 1.618
	Pinocembrin	2.000 50.000 100.000	98.142 ± 2.285 96.251 ± 1.169 93.291 ± 1.108	2.328 1.214 1.188	92.255 ± 3.320 96.911 ± 2.216 97.388 ± 0.619	3.598 2.287 0.636

**Table 5 molecules-27-03245-t005:** Intra-day, inter-day precision and accuracy of the four analytes in plasma and prostate (*n* = 3).

Sample	Component	Concentration (μg·mL^−1^/μg·g^−1^)	Intra-Day			Inter-Day		
			Mean ± S.D. (μg·mL^−1^/μg·g^−1^)	Accuracy (%, RE)	Precision (%, RSD)	Mean ± S.D. (μg·mL^−1^/μg·g^−1^)	Accuracy	Precision
							(%, RE)	(%, RSD)
Plasma	Puerarin	0.500	0.480 ± 0.03	−0.042	6.250	0.477 ± 0.047	−0.049	9.914
		10.000	10.493 ± 0.92	0.047	8.770	11.9 ± 0.436	0.160	3.663
		50.000	49.033 ± 2.957	−0.020	6.030	48.2 ± 1.179	−0.037	2.446
	Luteolin	0.200	0.233 ± 0.045	0.143	19.325	0.213 ± 0.026	0.061	12.034
		2.000	2.273 ± 0.118	0.120	5.211	2.137 ± 0.222	0.064	10.385
		5.000	4.877 ± 0.172	−0.025	3.530	5.217 ± 0.259	0.042	4.963
	Kaempferol	0.100	0.126 ± 0.039	0.206	18.615	0.127 ± 0.003	0.215	2.399
		0.500	0.523 ± 0.023	0.045	4.413	0.49 ± 0.012	−0.020	2.355
		2.000	1.973 ± 0.012	−0.014	0.585	2.05 ± 0.07	0.024	3.415
	Pinocembrin	0.100	0.137 ± 0.012	0.272	8.778	0.94 ± 0.044	0.894	4.637
		0.500	5.493 ± 0.224	0.909	4.072	0.497 ± 0.032	−0.007	6.472
		2.000	2.127 ± 0.061	0.060	2.873	2.203 ± 0.085	0.092	3.860
Prostate	Puerarin	0.100 1.000 5.000	0.088 ± 0.004 1.073 ± 0.142 4.973 ± 0.125	−0.132 0.068 −0.005	4.286 13.220 2.514	0.107 ± 0.014 1.070 ± 0.123 4.873 ± 0.134	0.068 0.065 −0.026	13.220 11.484 2.756
	Luteolin	0.200 5.000 10.000	0.197 ± 0.011 5.330 ± 0.085 9.490 ± 0.070	−0.014 0.062 −0.054	5.651 1.603 0.738	0.192 ± 0.003 4.590 ± 0.087 11.100 ± 0.700	−0.042 −0.089 0.099	1.378 1.899 6.306
	Kaempferol	2.000 100.000 200.000	2.300 ± 0.200 96.533 ± 2.658 203.467 ± 0.503	0.130 −0.036 0.017	8.696 2.753 0.247	2.000 ± 0.087 102.700 ± 3.161 202.633 ± 2.303	0.000 0.026 0.013	4.359 3.078 1.136
	Pinocembrin	2.000 50.000 100.000	2.087 ± 0.078 50.867 ± 1.484 103.867 ± 1.185	0.042 0.017 0.037	3.722 2.918 1.141	1.940 ± 0.044 49.933 ± 3.166 102.900 ± 0.608	−0.031 −0.001 0.028	2.247 6.340 0.591

**Table 6 molecules-27-03245-t006:** Stability test of four analytes in plasma and prostate (*n* = 3).

Sample	Component	Concentration (μg·mL^−1^/μg·g^−1^)	24 h at Room Temperature Freeze-Thaw Cycles					
			Mean ± S.D.	Accuracy	Precision	Mean ± S.D.	Accuracy	Precision
			(μg·mL^−1^/μg·g^−1^)	(%, RE)	(%, RSD)	(μg·mL^−1^/μg·g^−1^)	(%, RE)	(%, RSD)
Plasma	Puerarin	0.500	0.493 ± 0.015	−0.014	3.096	0.517 ± 0.01	0.033	2.019
		10.000	10.217 ± 0.07	0.021	0.687	10.333 ± 0.902	0.032	8.728
		50.000	49.967 ± 1.159	−0.001	2.320	50.667 ± 1.266	0.013	2.499
	Luteolin	0.200	0.24 ± 0.026	0.167	11.024	0.2 ± 0.018	0.000	8.789
		2.000	2.13 ± 0.0171	0.061	8.023	2.29 ± 0.098	0.127	4.301
		5.000	4.903 ± 0.091	−0.020	1.851	4.927 ± 0.055	−0.015	1.118
	Kaempferol	0.100	0.102 ± 0.015	0.023	14.959	0.088 ± 0.004	−0.131	3.960
		0.500	0.473 ± 0.035	−0.056	7.419	0.511 ± 0.019	0.022	3.624
		2.000	2.2 ± 0.056	0.091	2.531	2.047 ± 0.144	0.023	7.018
	Pinocembrin	0.100	0.117 ± 0.012	0.143	9.897	0.103 ± 0.015	0.032	14.495
		0.500	0.51 ± 0.02	0.020	3.922	0.503 ± 0.021	0.007	4.136
		2.000	2.01 ± 0.082	0.005	4.072	1.920 ± 0.026	−0.042	1.378
Prostate	Puerarin	0.100 1.000 5.000	0.973 ± 0.049 1.097 ± 0.138 4.940 ± 0.056	0.897 0.088 −0.012	5.068 12.580 1.127	0.100 ± 0.005 0.973 ± 0.042 5.057 ± 0.224	0.000 −0.027 0.011	5.292 4.277 4.423
	Luteolin	0.200 5.000 10.000	0.194 ± 0.005 4.963 ± 0.118 9.900 ± 0.458	−0.033 −0.007 −0.010	2.328 2.387 4.629	0.213 ± 0.031 4.833 ± 0.058 10.433 ± 0.551	0.062 −0.034 0.042	14.321 1.195 5.279
	Kaempferol	2.000 100.000 200.000	2.000 ± 0.265 99.133 ± 3.557 201.667 ± 3.215	0.000 −0.009 0.008	13.229 3.588 1.594	1.833 ± 0.058 96.667 ± 2.082 202.333 ± 2.082	−0.091 −0.034 0.012	3.149 2.153 1.029
	Pinocembrin	2.000 50.000 100.000	1.923 ± 0.076 51.000 ± 3.000 102.000 ± 2.646	−0.040 0.020 0.020	3.971 5.882 2.594	1.867 ± 0.153 51.333 ± 1.528 102.333 ± 2.887	−0.071 0.026 0.023	8.183 2.976 2.821

**Table 7 molecules-27-03245-t007:** Mean pharmacokinetic parameters for four analytes in rat plasma after oral administration of the aerial parts of *G. uralensis* extract at 31 days (*n* = 6).

Parameters	t_1/2_	T_max_	C_max_	AUC_0__–__t_	AUC_0__–__∞_	MRT_0__–__t_	MRT_0__–__∞_
	(h)	(h)	(ng/mL)	(h×ng/mL)	(h×ng/mL)	(h)	(h)
Puerarin	6.43 ± 0.20	0.50 ± 0.04	23.76 ± 1.05	238.05 ± 23.35	241.86 ± 2.45	14.40 ± 1.21	15.08 ± 1.71
Luteolin	31.08 ± 1.17	0.87 ± 0.05	185 ± 0.12	35.01 ± 0.81	51.63 ± 1.98	20.74 ± 1.91	43.95 ± 1.72
Kaempferol	18.98 ± 1.46	4.00 ± 0.17	1.27 ± 0.06	18.75 ± 1.18	23.79 ± 0.86	20.12 ± 0.84	31.82 ± 1.22
Pinocembrin	13.18 ± 0.72	1.50 ± 0.05	1.18 ± 0.06	21.78 ± 0.73	23.84 ± 1.29	19.12 ± 0.82	23.26 ± 0.83

**Table 8 molecules-27-03245-t008:** The specific parameters of analytes.

Name	Q1	Q3	DP	CE
Puerarin	415.1	267.1	−150	−46
Kaempferol	285.1	117	−150	−54
Luteolin	285	133.1	−135	−46
Pinocembrin	255	213	−120	−27
NEG-IS-NI	417	122	−100	−30

## Data Availability

The data presented in this study are available in [Pharmacokinetics, prostate distribution and metabolic characteristics of four representative flavones after oral administration of the aerial part of *Glycyrrhiza uralensis* in rats].

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
