# Peer review of "Pharmacokinetics, Prostate Distribution and Metabolic Characteristics of Four Representative Flavones after Oral Administration of the Aerial Part of Glycyrrhiza uralensis in Rats"

_molecules, 2022, doi:10.3390/molecules27103245_

Round 1
Reviewer 1 Report
the biological sample is so weak the spectral has not shown the proper setting From the overall analysis (Figure 3), the number of flavonoid glycosides in plasma, 104
prostate and urine was more than that of flavonoid aglycones, and the number of flavo- 105
noid glycosides and aglycones in faeces was the same. There are more flavonoid glycosides 106
in prostate and feces and they are mainly composed of flavonoid glycosides and flavonol 107
glycosides simples sentence are so weak data is not properly arranged formula are incorrect and so many errors were present
Reviewer 2 Report
- the quality control of the original plant extract should be performed and reported including the quantitative analysis of the major flavonoids (especially, the 4 compounds of interest). The HPLC chromatogram should be shown too.
- limit of detection should be added into the method validation section.
- please check page 18, line 324-325 "...For example, baicalin is not easy to be absorbed into blood. It is first hydrolyzed into baicalein in vivo, and then baicalein is re-generated by glucuronidation to play a role...". Is it baicalin or baicalein that is re-generated by glucuronidation to play a role.
- please check page 26, line 703-704 "In this study, we first investigated the metabolism of puerarin, kaempferol, luteolin and chaulmicin in the rat prostate..." Is it chaulmicin or pinocembrin ?
- The dose of this plant extract in human especially, for the treatment of CNP should be mentioned and discussed with the dose used in this animal experiment and the possible toxicity.
Author Response
Response to Reviewer 3 Comments
Point 1: Add the common name beside the scientific name.
Response 1: A generic name has been added after the scientific name appears for the first time in the text. “Glycyrrhiza uralensis Fisch. (Liquorice, G. uralensis)”.
Point 2: Identify this abbreviation at its first mention.
Response 2: Add in the text:[M-H-SO3]- (SO3-sulfate group);[M-H-CH3]- (CH3-methyl group)
Point 3: Insert the reference number
Response 3: The corresponding reference numbers have been added in the text.